# Transport of N-CD and Pre-Sorbed Pb in Saturated Porous Media

**DOI:** 10.3390/molecules25235518

**Published:** 2020-11-25

**Authors:** Salahaddin Kamrani, Vahab Amiri, Mosleh Kamrani, Mohammed Baalousha

**Affiliations:** 1Deputy for Technology Innovation and Commercialization Development, VPST, Tehran 1991745681, Iran; 2Department of Applied Geology, Faculty of Earth Sciences, Kharazmi University, Tehran 1571914911, Iran; 3Department of Geology, Faculty of Science, Yazd University, Yazd 89195741, Iran; v.amiri@yazd.ac.ir; 4Department of Chemical Engineering, Faculty of Engineering, University of Kurdistan, Sanandaj 6617715175, Iran; mosleh.kamrani@gmail.com; 5Center for Environmental Nanoscience and Risk, Arnold School of Public Health, University of South Carolina, Columbia, SC 29201, USA; 6Department of Environmental Health Sciences, Arnold School of Public Health, University of South Carolina, Columbia, SC 29201, USA

**Keywords:** nitrogen functional groups carbon dot, Pb remobilization, quartz sand column, adsorption affinity, transport experiment

## Abstract

Carbon dots (CDs) are a new type of nanomaterials of the carbon family with unique characteristics, such as their small size (e.g., <10 nm), high water solubility, low toxicity, and high metal affinity. Modification of CDs by Nitrogen functional groups (N-CDs) enhances their metal adsorption capacity. This study investigated the influences of pH (4, 6, and 9), ionic strength (1, 50, and 100 mM), and cation valency (Na^+^ and Ca^2+^) on the competitive adsorption of Pb to quartz and N-CD surfaces, the transport and retention of N-CDs in saturated porous media, and the capacity of N-CDs to mobilize pre-adsorbed Pb in quartz columns. Pb adsorption was higher on N-CDs than on quartz surfaces and decreased with increases in ionic strength (IS) and divalent cations (Ca^2+^) concentration. N-CD mobility in quartz columns was highest at pH of 9- and 1-mM monovalent cations (Na^+^) and decreased with decreases in pH and increases in ionic strength and ion valency. N-CDs mobilized pre-adsorbed Pb from quartz due to the higher adsorption affinity of Pb to N-CD than to quartz surfaces. These findings provide valuable insights into the transport, retention, and risk assessment of lead in the presence of carbon-based engineered nanoparticles.

## 1. Introduction

Carbon dots (CDs), a new member of the carbon-based engineered nanoparticles, are widely used in many applications, including sensors, cell imaging, photocatalysis, metal detection, organophosphate pesticide detection, light-emitting devices, energy storage devices, and organic photovoltaics [1,2,3]. CDs have important properties, such as high surface area and high metal affinity and sorption capacity, due to the abundance of surface functional groups, such as carboxylic, phenolic, and hydroxyl groups [4]. Functional groups on the surfaces of carbon nanoparticle play important roles in the removal of heavy metals from aqueous media [5]. The interactions between surface functional groups and the heavy metals can directly or indirectly affect the adsorption mechanisms, such as electrostatic interaction, surface complexation, ion exchange, physical adsorption, and precipitation [6].

Different chemical surface modification methods, such as oxidation, nitrogenation, and sulfuration, have been implemented to enhance the surface functionalization of CDs and, consequently, promote the adsorption of heavy metals [7]. Nitrogenation is a widely used approach to generate nitrogen-containing functional groups (e.g., -NH_2_, -NH, -C=N, and -C-N) on the surfaces of carbon-based materials [1]. The introduction of nitrogen onto carbon-based nanoparticles can significantly enhance the polarity of its surface and thus increase its specific interaction with polar adsorbates [8].

Several studies investigated the transport of carbon-based nanoparticles in sand columns under saturated conditions [9]. These studies demonstrated that solution chemistry (e.g., pH, ionic strength (IS) and composition, and ion valency) and soil properties (e.g., grain size, charge behavior, etc.) impact the aggregation and transport of nanoparticles in soils. For example, Yu et al. [10] observed an increased transport and mobility of Graphene quantum dots (GQDs) at higher pH and lower IS. They also observed that higher input concentration of GQDs slightly increased the GQDs retention due to the ripening process. Li et al. [11] compared the transport behavior of N-doped graphene (NG) and Graphene oxide (GOs) in quartz column and demonstrated that GOs were more mobile than NG in quartz column and that the mobility of NG decreased with increases in IS and decreases in quartz media grain size. Additionally, the facilitated transport of heavy metals by carbon-based nanoparticles have received significant attention recently. For instance, Zhou et al. [12] investigated the effects of GO concentration and IS on the transport of Cu^2+^ in the sand column. They reported increases in Cu^2+^ transport with decreases in IS and increases in GO concentrations. Additionally, GO have been shown to remobilize pre-sorbed Pb^2+^ and Cd^2+^ in porous media [13]. Kamrani et al. [14,15] observed increased mobility of CDs in both saturated and unsaturated porous media with decreases in IS and increases in pH and collector grain size. They also found that CDs remobilize pre-adsorbed Cu and Pb in saturated porous media, attributed to CDs’ high sorption capacity. However, studies on the impact of environmental conditions on the transport, retention of N-CDs, and metal mobilization by N-CDs are lacking.

The overall objective of this study is to investigate the impact of pH, ionic strength, and valency on the fate and transport of N-CDs in saturated porous media. Quartz with 0.2–0.5 mm grain size was used as an experimental porous medium. The specific objectives of this study are to (1) determine the influences of pH and ionic strength and valency on the transport of N-CDs in saturated quartz columns; (2) determine the impact of N-functionalization of CDs on their transport behavior in quartz columns; and (3) investigate efficacy of N-CDs to remobilize pre-adsorbed lead from quartz columns.

## 2. Results and Discussion

### 2.1. Properties of N-CDs

The FT-IR spectra of the N-CDs and CDs display similar peaks that were assigned to surface functional groups, such as C-H, C=O, and O-H (Figure 1A). However, the FT-IR spectrum of N-CDs displays peaks that were not present in the CD spectrum including N bands; N-H bonds at 3277 cm^−1^, C=N bonds at 1663 cm^−1^, and C-N and aromatic C-NH bonds at 1399 and 1285, respectively [16]. The oxygen-containing functional groups (carboxylic and hydroxylic) and amino functional groups on the surfaces of N-CDs are responsible for the high metal ion sorption on N-CDs’surfaces [17].

Raman scattering spectra reveal the structural features of the carbon atoms within N-CDs and CDs. The Raman spectra shows two prominent peaks of the D and G band at 1365 and 1585 cm^−1^ (Figure 1B,C). The G band is a result of in-plane vibrations of sp^2^ bonded carbon atoms, whereas the D band is due to the out of plane vibrations attributed to the presence of structural defects or disorder in carbon-based nanomaterials [18,19]. The ratio of the intensity of D/G (*I*_D_/*I*_G_) is a measure of disorder of defects present in carbon-based nanomaterials. The *I*_D_/*I*_G_ of the N-CDs (0.94) is higher than the CDs (0.86), indicating an increased disorder defects in N-CDs compared to CDs [20,21].

The Zeta-potential of N-CDs and CDs under similar experimental conditions were negatively charged (Table 1). The Zeta-potential of N-CDs was slightly higher than that of CDs under the same experimental conditions (Table 1), which can be ascribed to the amino-functional groups on N-CD surface [22]. The Zeta-potential of N-CDs increased with increases in IS and decreases in pH. The hydrodynamic diameter of N-CDs was larger than that of CDs under the same experimental conditions, which is attributed to higher N-CD aggregation than CD under the same experimental conditions. The UV-vis absorption spectra of CDs and N-CDs exhibited noticeable sharp absorption peaks at 335 nm (Figure 2A), which was ascribed to the n-π^*^ electronic transition of CDs/N-CDs. N-CDs demonstrated higher absorbance than CDs, which can be attributed to the amino-functional groups attached to the surfaces of the N-CDs [23]. The UV-vis absorbance of N-CDs during 300 min after synthesis was nearly constant, indicating that N-CDs remained stable and did not undergo aggregation and, subsequently, sedimentation (Figure 2B). The UV-vis absorbance at 335 nm varied slightly with changes in pH (Figure 2C). This phenomenon is likely due to the protonation‒deprotonation of the functional groups on N-CD surfaces [24].

The absorbance of N-CDs decreased slightly with increases in NaCl/CaCl_2_ concentrations (Figure 2D), likely due to alteration of surface plasmon resonance at higher ionic strengths due to Na^+^ and Ca^2+^ sorption or due to N-CD aggregation [25]. Therefore, a correction was applied to convert the measured absorbances to the corresponding CD and N-CD concentrations at different pH and IS conditions in the transport experiments.

### 2.2. N-CD Aggregation and N-CD-Quartz Interactions

At the same pH, the hydrodynamic diameter of N-CDs increased with increases in IS (Table 1), which can be attributed to N-CD aggregation due to the decrease in zeta potential and the interaction energy barrier (Appendix A). Additionally, the hydrodynamic diameter of the N-CDs was larger in the presence of Ca^2+^ than Na^+^ at the same molar concentration, due the higher efficiency of Ca^2+^ in screening and neutralizing the surface charge of the N-CD than Na^+^ in good agreement with Chulze-Hardy rule and to bridging of N-CDs by Ca^2+^ [26]. The zeta potential of N-CDs was lower in Ca^2+^ solutions than in Na^+^ solutions (Table 1), and, subsequently, the interaction energy barrier between N-CDs was lower (e.g., 19.1–0.9 kT in 1–100 mM) in the presence of Ca^2+^ than (e.g., 27.1–13.2 kT in 1–100 mM) in the presence of Na^+^ (Appendix A).

The total repulsive potentials between N-CDs and quartz surfaces decreased with increases in IS. The DLVO calculations indicate a distinct decrease in the secondary energy minimum between N-CDs and quartz and a decrease in the distance of energy minimum position with the increased IS, especially in CaCl_2_ solutions (Appendix A). Therefore, the N-CDs are likely to occupy reversible adsorption sites on the quartz surfaces at the secondary energy minima [27,28].

At the same ionic strength, the hydrodynamic diameter of N-CDs increased with decreases in pH, which can be attributed to decreases in N-CD zeta potential due to protonation of carboxylic groups at lower pHs (Table 1) and the subsequent decrease in the interaction energy barrier between N-CDs (Appendix A). Similarly, the primary interaction energy barrier between N-CDs and quartz decreased with the decreases in pH. The secondary minimum position between N-CDs (Appendix A) and between N-CDs and quartz (Appendix A) is nearly constant at different pHs. For all the pH condition in this study, the energy barriers for N-CDs aggregation and adsorption on quartz are high (>11.2 kT, Appendix A), and the total repulsive potentials remain positive for long separation distances (<100 nm (Appendix A)), indicating that, under the experimental conditions used in this study, the interaction between N-CDs and quartz is unfavorable for aggregation and irreversible adsorption [29].

### 2.3. Pb Adsorption on N-CD and Quartz Surface

In a mixture of 10 g quartz and 50 mg L^−1^ N-CDs, higher concentrations of Pb were adsorbed on N-CDs than on quartz surfaces under all experimental conditions (Figure 3A,B), indicating higher affinity of Pb to N-CDs than to quartz surface. For example, at pH6 and 1 mM IS, N-CDs adsorbed 55.1% and 47.8% of Pb in NaCl and CaCl_2_ solutions, respectively, and quartz surface adsorbed about 12% in both NaCl and CaCl_2_ solutions (Figure 3). Higher adsorption of N-CDs is attributed to high binding affinity of N-CDs functional groups with vacant d-orbitals of Pb ions [30].

The adsorption of Pb on N-CD and quartz surfaces in acidic and alkaline media was lower than that under neutral pH. In acidic medium (pH 4), protonation of the functional groups on the N-CD/quartz surfaces results in a decrease in Pb adsorption on the N-CDs/quartz surfaces [31,32]. The decreased adsorption of Pb onto N-CD and quartz under alkaline conditions (pH 9) compared to neutral pHs can be explained by the formation of Pb hydroxide complexes (hydrocerrusite) [33] and precipitates [14,34].

Pb sorption on N-CD/quartz surfaces decreased with the increase in concentration of NaCl and CaCl_2_ solutions (Figure 3B). This decrease can be attributed to Pb^2+^ competition with cations (Ca^2+^ and Na^+^) for adsorption sites on N-CD/quartz surfaces. Furthermore, the formation of soluble lead chloride species can prevent the adsorption of Pb^2+^ ions on the N-CD/quartz surfaces at high chloride concentrations [15].

### 2.4. Impact of N-Functionalization on the CD Transport and Pb Transport

Figure 4 presents the breakthrough curves (BTCs) of N-CDs, CDs, Pb, and released Pb (r-Pb) under two different cation types (NaCl and CaCl_2_) at pH 6. The BTCs are plotted as normalized effluent concentrations (C/C_0_) versus pore volumes (PVs). The first breakthrough of N-CDs/CDs and the adsorbed Pb in the effluent were at 0.75 and 1.05 PVs, respectively, for both NaCl and CaCl_2_.

The BTCs of N-CDs and CDs, under the same experimental conditions, were similar in shape (symmetric), but N-CD recovery was lower than CD recovery under the same conditions (Table 2), which can be attributed to the lower interaction energy barrier between quartz and N-CDs than that between quartz and CD (Appendix A) [35]. Additionally, the physical straining of N-CDs could be higher than that of CDs under the same experimental conditions due to the increased N-CD aggregate sizes compared to CD aggregate sizes under the same experimental conditions (Table 1). The effluent recoveries for these experiments are provided in Table 2.

In the absence of N-CD/CD injection, Pb started eluting at 1.95 PV under NaCl and CaCl_2_ conditions and most Pb (e.g., 93–95%) was retained in the quarts column (Table 2, Figure 4). The introduction of N-CDs/CDs resulted in the remobilization of the retained Pb, as indicated by the BTCs of Pb for both solutions (NaCl and CaCl_2_). In NaCl solution, the maximum normalized effluent concentrations (C/C_0_ max) of Pb increased from 0.08 to 0.88 for N-CDs and 0.08 to 0.75 for CDs. In CaCl_2_ solution, the C/C_0_ max of Pb increased from 0.05 to 0.73 for N-CDs and 0.08 to 0.61 for CDs. Furthermore, the C/C_0_ max of r-Pb after N-CD injection was higher than the C/C_0_ max of r-Pb after CD injection (Figure 4), which can be attributed to the higher affinity of Pb^2+^ to N-CD surfaces than to CD surfaces. This, in turn, can be attributed to the higher affinity of Pb^2+^ to the amino-functional groups on the N-CD surfaces [36] than to the oxygenated functional groups on the surfaces of CDs. Injection of N-CDs remobilized 56.9% and 53.4% of the retained Pb in NaCl and CaCl_2_ solutions (Table 2). On the other hand, the injection of CDs remobilized 50.6% and 46.6% of Pb in NaCl and CaCl_2_ solutions. The higher remobilization of Pb following N-CD injection than CD injection can be attributed to the higher affinity of Pb to N-CDs than to CDs, as discussed above.

### 2.5. Effects of Environmental Factors on N-CDs and r-Pb Transport

#### 2.5.1. Effects of Ionic Strength and Cation Type

Increases in IS from 1 to 100 mM did not impact the breakthrough arrival time of N-CDs, in both NaCl and CaCl_2_ solutions. The BTCs of N-CDs were flat at 1 mM IS and reached a C/C_0_ max of 1 (similar to the conservative tracer) in both NaCl and CaCl_2_ solutions (Figure 5). In both NaCl and CaCl_2_ solutions, increases in IS resulted in lower flat BTC plateau heights, indicating temporal constancy in the deposition rate during the transport experiment time [37]. The BTC plateau heights of N-CDs decreased from 1.02 to 0.77 and from 1.03 to 0.68 when the IS increased from 1 to 100 mM in NaCl and CaCl_2_, respectively, suggesting the increased retention of N-CDs in the quartz sand column with the increase in IS (Table 2). The plateau height of N-CDs was also lower in CaCl_2_ than in NaCl, indicating the higher retention of N-CDs in CaCl_2_ solutions than in NaCl solution at the same molar concentration, which is in good agreement with the DLVO theory calculation, as discussed above.

The plateau height (C/C_0_) of Pb decreased with the increased IS (Figure 5A,B). The elution time of r-Pb was delayed in CaCl_2_ compared to that in NaCl, which can be attributed to competition of Ca with Pb to N-CD surfaces, the aggregation of N-CDs in CaCl_2_, and thus N-CD straining in the quartz column, or to the attachment of N-CDs to quartz surfaces via van der Waals interactions [9]. The recovery of N-CDs and r-Pb decreased with increases in NaCl and CaCl_2_ concentrations (Table 2). Additionally, Pb recovery was lower in CaCl_2_ than in NaCl. Therefore, it can be concluded that divalent cations inhibited the transport of N-CDs/r-Pb to a higher extent than monovalent cations at the same molar concentrations [38]. For example, the effluent recoveries of N-CDs and r-Pb were 65.2% and 21.6% at 100 mM of NaCl and 48.5% and 11.3% at the same IS of CaCl_2_.

The BTC plateaus of r-Pb decreased in line with the general trend of BTCs of N-CD decreasing, indicating that Pb was adsorbed on N-CDs [39]. Under higher IS, the straining process was more pronounced due to the electrostatic double layer compression and subsequent N-CDs agglomeration [40,41,42], and the decrease in C/C_0_ of Pb is due to the decrease in the amount of N-CDs output.

#### 2.5.2. Effects of pH

The retention of N-CDs increased with decreases in pH (Figure 6A,B, Table 2). This trend is attributed to the increased protonation of the carboxylic, hydroxylic, and amino functional groups and the formation of positively charged sites on N-CD surfaces with the decrease in pH [43,44]. Thus, the magnitude of N-CDs decreased with decreases in pH resulting in decreases in the energy barrier between N-CDs and quartz (Appendix A).

After N-CD injection, the C/C_0_ of r-Pb increased with the increase in pH from 4 to 6 and then decreased with the increase in pH to 9 for both NaCl and CaCl_2_ solutions. The remobilized Pb (r-Pb) increased from 20.9% in NaCl and 8.7% in CaCl_2_ at pH 4 to 56.9% in NaCl and 53.4% in CaCl_2_ at pH 6 and then decreased to 25.5% NaCl and 13.1% in CaCl_2_ at pH 9 (Table 2). The low Pb recovery at pH 4 is likely due to retention of N-CDs in the quartz column, indicated by the lower C/C_0_ (Figure 6). On the other hand, the low Pb recovery at pH 9 can be attributed to the decreased Pb sorption on N-CD surfaces, as discussed above.

## 3. Material and Methods

### 3.1. Preparation of N-CDs

Carbon dots were synthesized from citric acid using a simple, one-step polymerization reaction method according to the method reported elsewhere [14]. To prepare the nitrogen-functionalized carbon dots (N-CDs), 4 g of citric acid (C_6_H_8_O_7_—Merck, Darmstadt, Germany) was mixed with 780 μL ethylenediamine (≥99%, Merck, Darmstadt, Germany) and 10 mL of deionized (DI) water in a 100 mL glass beaker. The resulting solution was hydrolyzed at 250 °C for 60 min, resulting in the formation of a red-brown foamy solid. The foamy solid was dissolved in double-distilled water and centrifuged (Allegra 64R—Beckman, Irving, TX, USA) at 12,000× *g* for 20 min to remove large particles. Then, the foamy solid was dialyzed against Millipore water using a dialysis membrane (MWCO of 3kD—Merck, Darmstadt, Germany) to remove dissolved species (e.g., <1 nm). Before each experiment, the stock suspension was diluted in the experimental electrolyte solution to the desired N-CDs/CDs concentration.

### 3.2. Characterization of N-CDs

The electrophoretic mobility and the z-average hydrodynamic diameter of N-CDs were measured at 25 °C under different pH and IS conditions using laser Doppler electrophoresis and dynamic light scattering (DLS), respectively (Zetaplus, Brookhaven, NY, USA). Zeta potential was calculated from the measured electrophoretic mobility using Henry’s function and Smoluchowski approximation [45]. Surface functional groups were determined by Fourier transform-infrared spectrometer (FT-IR) (Perkin-Elmer, RX1, KBr disks, Waltham, MA, USA). The N-CDs’ and CDs’ concentrations were determined by UV-vis at λ = 335 nm. Raman spectroscopy was performed on a MultiRam Raman spectrometer (Bruker, Billerica, MA, USA). Lead concentration was measured by Atomic absorption spectroscopy (AAS, Varian AA240 FS, Santa Clara, CA, USA).

### 3.3. Porous Media

Quartz (with ~99.38% SiO_2_ and 0.27 Fe_2_O_3_ based on X-Ray Fluorescence (XRF) analysis) was used as the experimental porous medium. The size of quartz grains varied between 0.2 and 0.5 mm with a mean grain diameter of 0.4 mm. The quartz was cleaned following the protocol described elsewhere [46]. In brief, the quartz grains were cleaned to remove organic and inorganic impurities by soaking in 70% HNO_3_ for 16 h, followed by rinsing with DI water until pH equilibrated, followed by sonication. The washing step with DI water was repeated until the turbidity of the supernatant was negligible as verified by UV/Vis at 335 nm. Lastly, the quartz grains were dried at 300 °C for at least 24 h. The zeta potential of the cleaned quartz was measured using Zeta sizer (BI-EKA, Brookhaven, Holtsville, NY, USA) in both NaCl and CaCl_2_ solutions under the experimental conditions.

### 3.4. Column Transport Experiments

Treated quartz was wet packed into an acrylic organic glass column (12 cm length × 2.5 cm inner diameter) with stainless mesh screens at both ends. Each column contained 112 ± 5 g of quartz sand and had an average porosity of 0.24 ± 0.01. Each solution was introduced to the column using a peristaltic pump (Shenchen pump, BT100K, Baoding, China) in the upward direction at a constant Darcy’s velocity of 3.075 mL·min^−1^. All column experiments were performed in three and five steps according to the method reported elsewhere [15]. Briefly, the three-step experiment included: (1) background flushing, (2) background and Pb/Tracer, and (3) background flushing. The five-step experiment included: (1) background flushing, (2) background and Pb, (3) background flushing, (4) background and N-CDs, and (5) background flushing. Effluent samples were collected in polypropylene tubes continuously by a fraction collector (BS–100A, Shanghai, China). The normalized effluent concentration (C/C_0_) for each transport experiment was plotted as a function of pore volumes. The experimental protocols of the column experiments are summarized in Table 1. In addition, a tracer solution (NaCl) was injected in the column under similar experimental conditions. All experiments were performed in triplicate, and all data are presented as the mean and standard deviation of the triplicate experiments.

### 3.5. Batch Experiments

Sorption of Pb to N-CDs and quartz grains was performed by mixing 50 mg L^−1^ N-CDs and 10 g of quartz with 10 mg L^−1^ Pb(Cl)_2_ solution for 120 min. Lead sorption to N-CDs was examined under different pHs (4, 6, and 9) and ISs (1, 50, and 100 mM) and compositions (NaCl and CaCl_2_). Quartz grains were separated with stainless mesh screens, and Pb sorbed on quartz was released by mixing quartz grains with 25 mL of 0.5% HCl and rotated for 20 min. Lead sorbed on N-CDs was separated from dissolved Pb by ultrafiltration at 3 kDa using centrifugal ultrafiltration units at 5000× *g* for 20 min. The concentration of Pb was measured by Atomic absorption spectroscopy (AAS, Varian AA240 FS, Santa Clara, CA, USA). Electrophoretic mobility and z-average hydrodynamic diameter of N-CDs in the experimental solution chemistries were quantified by laser Doppler electrophoresis and dynamic light scattering (DLS), respectively.

### 3.6. Derjaguin, Landau, Vervey, and Overbeek (DLVO) Theory

The total interaction potential between particles was calculated based on DLVO theory as the sum of attractive van der Waals (vdW) potential and repulsive electrostatic double layer (EDL) potential. Although other forces may influence the net interaction forces (e.g., hydration) [47], we maintain simplicity in this investigation by focusing on van der Waals and electric double layer forces [37]. DLVO calculations are presented in the Appendix A section.

## 4. Conclusions and Prospective

This study systemically investigated the transport, retention, and adsorption capacity of N-CDs in packed quartz columns under a range of environmentally relevant IS and pH conditions in both monovalent and divalent electrolytes. The mobility of N-CDs increased with decreases in IS and increases in pH. N-CDs displayed lower mobility than CDs in the quartz columns, but N-CDs displayed higher affinity to Pb adsorption than CDs. N-CDs remobilized pre-sorbed Pb under all experimental conditions investigated in this study. The optimal (e.g., highest) remobilization of pre-sorbed Pb occurred at pH 6. In contrast, the remobilization of pre-sorbed Pb was lower at pH 4 due to reduced N-CD mobility, as well as at pH 9, due to reduced Pb sorption on N-CDs. Remobilization of pre-sorbed Pb was lower in CaCl_2_ than in NaCl due to decreased zeta potential of N-CDs and quartz and, thus, increased N-CD aggregation and increased N-CD attachment to quartz. Overall, this study demonstrated that N-CDs could serve as an effective carrier for Pb at pH 6 and low IS.

Natural organic matter (NOM) is known to form surface coating on the surfaces of natural and engineered particles [48,49,50]; thus, NOM is likely to form a surface coating on the surfaces of CDs and quartz, which may affect the mobility of N-CDs in the quartz column. Additionally, NOM is rich with surface functional groups and is thus likely to compete with CDs and quartz for the sorption of Pb^2+^ ions. Subsequently, NOM is likely to influence the mobility of CDs and Pb^2+^ in quartz columns and the remobilization of pre-sorbed Pb^2+^ in quartz columns. Future studies are need to investigate the influences of natural organic matter (NOM) on the sorption of Pb on N-CD and quartz surfaces, as well as the impact of these interactions on Pb mobility in quartz columns.

## Figures and Tables

**Figure 1 molecules-25-05518-f001:**
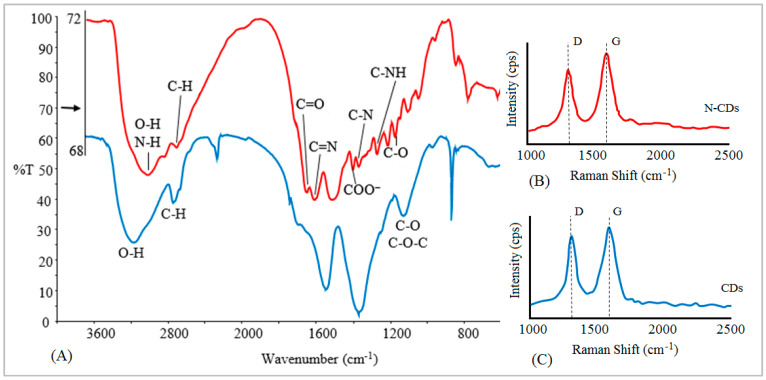
(**A**) Fourier transform-infrared (FT-IR) spectra of (red) nitrogen functionalized carbon dots (N-CDs) and (blue) nonfunctionalized carbon dots (CDs). (**B**) Raman spectra of N-CDs. (**C**) Raman spectra of CDs.

**Figure 2 molecules-25-05518-f002:**
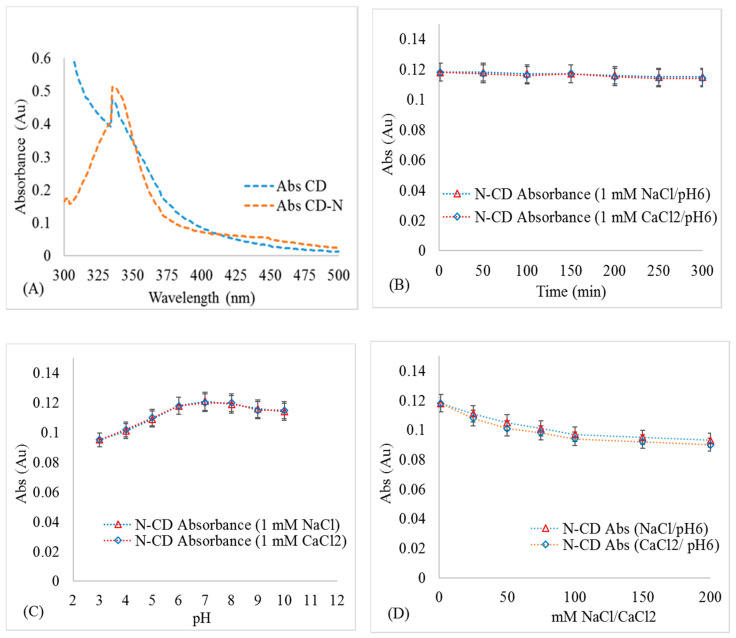
Stability of 50 mg L^−1^ N-CD: (**A**) UV-vis spectra of N-CD/CD at 200 mg L^−1^. (**B**) Absorbance at 335 nm at different times after synthesis; (**C**) change in the absorbance at 335 nm at different pH; and (**D**) at different IS.

**Figure 3 molecules-25-05518-f003:**
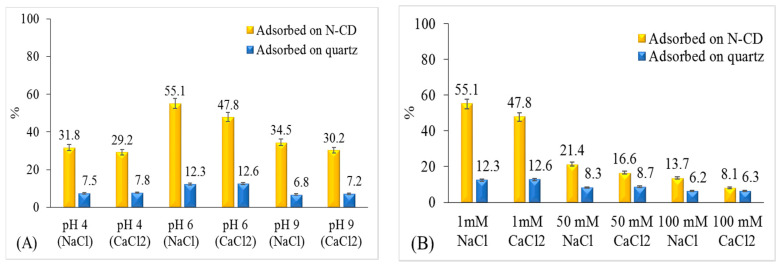
Batch experiments of Pb adsorption on N-CD and CD under different pH (**A**). Ionic strength (IS) (**B**). Ten grams quartz, 25 mL Deionized water, N-CD/CD: 50 mg.L^−1^, Cu: 10 mg. L^−1^, and equilibrium time 120 min, background solution containing: NaCl (1 mM), NaHCO_3_ (1 mM).

**Figure 4 molecules-25-05518-f004:**
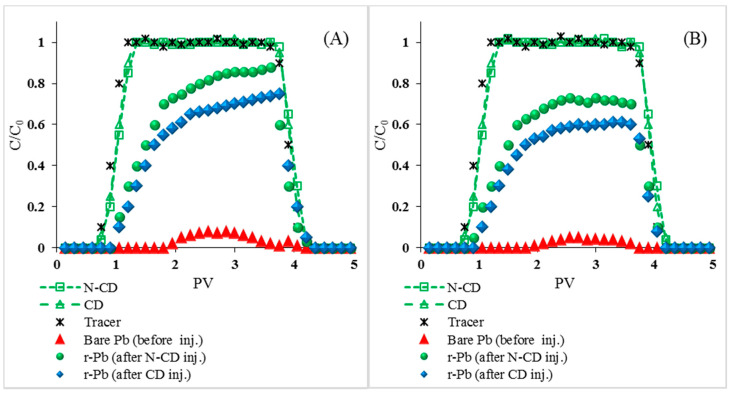
Breakthrough curve (BTC) of N-CD, CD, Bare Pb, released Pb (r-Pb) and tracer: NaCl (**A**), CaCl_2_ (**B**). All experiments were performed in C_0_ (N-CD/CD): 50 mg·L^−1^, C_0_ (Pb): 10 mg·L^−1^, pH6, background solution containing: 1 mM NaCl/CaCl_2_, NaHCO_3_ (1 mM), 1 rpm (3.075 mL·min^−1^) flow rate.

**Figure 5 molecules-25-05518-f005:**
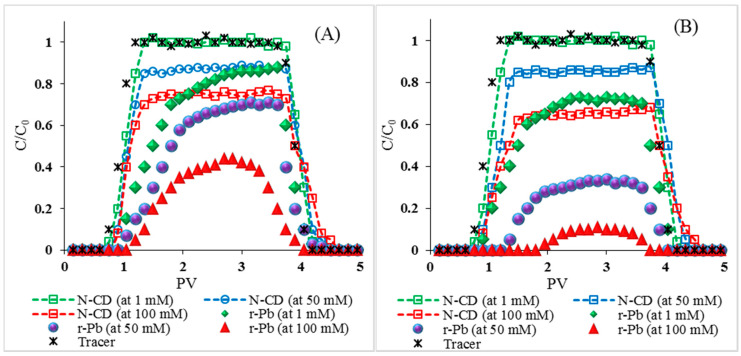
Breakthrough curves (BTCs) of N-CD and released Pb (r-Pb) at different ionic strength: NaCl (**A**), CaCl_2_ (**B**). All experiments were performed in C_0_ (N-CD/CD): 50 mg·L^−1^, C_0_ (Pb): 10 mg·L^−1^, pH 6, background solution containing: 1 mM NaCl/CaCl_2_, NaHCO_3_ (1 mM), 1 rpm (3.075 mL·min^−1^) flow rate.

**Figure 6 molecules-25-05518-f006:**
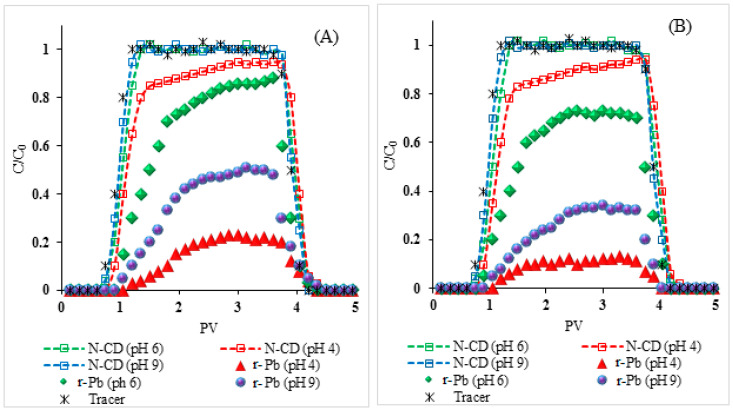
Breakthrough curves (BTCs) of N-CD and released Pb (r-Pb) at different pH. NaCl (**A**), CaCl_2_ (**B**). All experiments were performed in C_0_ (N-CD/CD): 50 mg·L^−1^, C_0_ (Pb): 10 mg·L^−1^, pH6, background solution containing: 1 mM NaCl/CaCl_2_, NaHCO_3_ (1 mM), 1 rpm (3.075 mL·min^−1^) flow rate.

**Table 1 molecules-25-05518-t001:** Zeta potential for nitrogen functionalized carbon dots (N-CDs) and sand grains at different pH and iconic strength (IS) solutions. N-CD/CD concentration: 50 mg·L^−1^.

pH	IS (mM NaCl)	IS (mM CaCl_2_)	N-CDs Zeta Potential-mV (±2.5)	CDs Zeta Potential-mV (±2.5)	N-CD Hydrodynamic Diameter-nm (±5)	CD Hydrodynamic Diameter-nm (±5)	Quartz Grains Zeta Potential-mV (±2.5)
4	1	0	−21.6	−24.2	44.3	39.3	−40
6	1	0	−26.2	−29.6	30.7	24.6	−65
9	1	0	−33.5	−38.6	19.4	17.5	−80
6	50	0	−21.5	−25.1	67.1	64.9	−51.6
6	100	0	−15.8	−19.2	75.2	71.3	−44.6
4	0	1	−14.5	−16.9	51.1	45.4	−37.5
9	0	1	−28.2	−33.8	25.5	21.3	−75.3
6	0	1	−21	−26.3	36.3	30.8	−61.2
6	0	50	−10.2	−12	121.5	113.6	−45.1
6	0	100	−8.6	−9.1	136.6	128.2	−37.2

**Table 2 molecules-25-05518-t002:** N-CDs, CDs, and Pb recovered in column experiments at different condition (Standard Deviation (STD) < 5%).

Materials	Exp. Conditions	Recovered Nanoparticles (%)	Total Effluent Pb (%)	Recovered Pb after Nanoparticles Injection (%)
N-CD (50mg/L, Pb: 10 mg/L)	1 mM NaCl, pH 6	93.3	63.1	56.9
	1 mM CaCl_2_, pH 6	91.1	58.2	53.4
	50 mM NaCl, pH 6	76.4	39.7	33.5
	50 mM CaCl_2_, pH 6	62.7	33.1	28.3
	100 mM NaCl, pH 6	65.2	27.8	21.6
	100 mM CaCl_2_, pH 6	48.5	16.1	11.3
	1 mM NaCl, pH 4	82.6	27.1	20.9
	1 mM CaCl_2_, pH 4	74.4	13.5	8.7
	1 mM NaCl, pH 9	98.2	31.7	25.5
	1 mM CaCl_2_, pH 9	98.4	17.9	13.1
Carbon Dot (CD: 50 mg/L)	1 mM NaCl, Ph 6	96.1	56.8	50.6
	1 mM CaCl_2_, pH 6	94.6	51.4	46.6
Effluent Pb (before nanoparticles injection)	1 mM NaCl, pH 6	-	6.2	-
Effluent Pb (before nanoparticles injection)	1 mM CaCl_2_, pH 6	-	4.8	-

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
