# Peer review of "Transport of N-CD and Pre-Sorbed Pb in Saturated Porous Media"

_molecules, 2020, doi:10.3390/molecules25235518_

Round 1

Reviewer 1 Report

This is an interesting and well written paper. This reviewer has the following suggestions/comments

1. In the introduction the authors mention chemical surface modification have been used to enhance the surface functionalization of CDs and promote the adsorption of heavy metals. They also mention studies investigated the transport of carbon-based nanoparticles in sand and these studies demonstrated that solution chemistry and soil properties impact the aggregation and transport of nanoparticles in soils.

Have the authors also considered how CDs may interact with natural aquatic colloids composed of humic material? These naturally occurring nanoparticles are known to also be involved in the adsorption of heavy metals and other toxic materials. There has been a number of papers published studying these types of materials and how they not only form nanoparticles but also films. Could functionalised CDs also remove this material from waterways and soils? Some papers of note in this area are as follows

- Characterization of freshwater natural aquatic colloids by atomic force microscopy (AFM), JR Lead, D Muirhead, CT Gibson, Environmental science & technology 39 (18), 6930-6936, 2005

- Quantifying the dimensions of nanoscale organic surface layers in natural waters, CT Gibson, IJ Turner, CJ Roberts, JR Lead, Environmental science & technology 41 (4), 1339-1344, 45, 2007

- Aquatic colloids and nanoparticles: current knowledge and future trends, JR Lead, KJ Wilkinson, Environmental Chemistry 3 (3), 159-171, 2006

Please note the authors are under no obligation to cite the above articles they are only a suggestion

  1. Microplastics and nanoplastics are now being investigated as a huge problem in terms of environmental waste with obvious health effects. Can CDs be used in any way to “capture” and negate this material. Please discuss.

  1. Did the authors consider using electron microscopy to characterise the CDs? While DLS is a well known technique some problems can occur with its use particularly around aggregation. SEM and/or TEM may have provided some supporting data to confirm the DLS results.

  1. Another technique that can be sued to characterise CDs is Raman microscopy and can give important information on the disorder or defective nature of the CDs. This is particularly relevant before and after functionalisation. Some mention of this technique and its potential in characterising these types of materials would be prudent to mention. The following paper is an example where Raman was applied to CD and also shows an interesting application for CD nanocomposites in fingermark detection

Non‐toxic luminescent carbon dot/poly (dimethylacrylamide) nanocomposite reagent for latent fingermark detection synthesized via surface initiated reversible addition fragmentation chain transfer polymerization, J Dilag, H Kobus, Y Yu, CT Gibson, AV Ellis, Polymer International 64 (7), 884-891, 2015.  

  Please note the authors are under no obligation to cite the above article it is only a suggestion.

Author Response

Dear Editor,

We thank the reviewers for their constructive feedback, which helped us to improve the manuscript. Below, we address the reviewers’ comments one by one and discuss the subsequent modifications to the manuscript. All requested changes have been marked in blue.

Reviewer 1

This is an interesting and well written paper.

Our response: we thank the reviewer for the overall positive feedback.

This reviewer has the following suggestions/comments

  1. In the introduction the authors mention chemical surface modification have been used to enhance the surface functionalization of CDs and promote the adsorption of heavy metals. They also mention studies investigated the transport of carbon-based nanoparticles in sand and these studies demonstrated that solution chemistry and soil properties impact the aggregation and transport of nanoparticles in soils.

Have the authors also considered how CDs may interact with natural aquatic colloids composed of humic material? These naturally occurring nanoparticles are known to also be involved in the adsorption of heavy metals and other toxic materials. There has been a number of papers published studying these types of materials and how they not only form nanoparticles but also films. Could functionalised CDs also remove this material from waterways and soils? Some papers of note in this area are as follows

- Characterization of freshwater natural aquatic colloids by atomic force microscopy (AFM), JR Lead, D Muirhead, CT Gibson, Environmental science & technology 39 (18), 6930-6936, 2005

- Quantifying the dimensions of nanoscale organic surface layers in natural waters, CT Gibson, IJ Turner, CJ Roberts, JR Lead, Environmental science & technology 41 (4), 1339-1344, 45, 2007

- Aquatic colloids and nanoparticles: current knowledge and future trends, JR Lead, KJ Wilkinson, Environmental Chemistry 3 (3), 159-171, 2006

 Please note the authors are under no obligation to cite the above articles they are only a suggestion

Our response: We thank the reviewer for pointing this out. Natural organic matter is indeed an important component of natural systems and their effect on Pb interactions with CDs and quartz, and the subsequent mobilization of Pb in quartz columns will be investigated in future studies. The following paragraph was added to the conclusions section “Natural organic matter is known to form surface coating on the surfaces of natural and engineered particles (Lead et al. 2005; Lead and Wilkinson. 2006; Gibson et al. 2007) and thus NOM is likely to form a surface coating on the surfaces of CDs and quartz, which may affect the mobility of N-CDs in the quartz column. Additionally, NOM is rich with surface functional groups and is thus likely to compete with CDs and quartz for the sorption of Pb2+ ions. Subsequently, NOM is likely to influence the mobility of CDs and Pb2+ in quartz columns and the remobilization of pre-sorbed Pb2+ in quartz columns. Future studies are need to investigate the influences of natural organic matter on the sorption of Pb on N-CD and quartz surfaces, and the impact of these interactions on Pb mobility in quartz columns.” See page 11, 2nd paragraph.

  1. Microplastics and nanoplastics are now being investigated as a huge problem in terms of environmental waste with obvious health effects. Can CDs be used in any way to “capture” and negate this material. Please discuss.

Our response: Considering the size of CDs (< 10 nm) and the size of microplastics (< 5 mm) and nanoplastics (< 100 nm, or <1000 nm depending on the definition), it is likely that CD’s will sorb on the surface of microplastics and nanoplastics and thus, it is likely that microplastics and nanoplastics may act as a vector for the transport of CDs. Therefore, we do not expect CDs to be used in anyway to capture and negate microplastics and nanoplastics. Although this is an interesting question, it beyond the scope of this study. No changes were made in response to this comment.

  1. Did the authors consider using electron microscopy to characterise the CDs? While DLS is a well-known technique some problems can occur with its use particularly around aggregation. SEM and/or TEM may have provided some supporting data to confirm the DLS results.

Our response: TEM analysis of the synthesized CDs were presented in previous publications (Kamrani et al., Wat. Res. 2018) and not presented here. No changes were made in response to this comment.

Reference: Kamrani S, Rezaei S, Kord M, Baalousha M. Transport and retention of carbon dots (CDs) in saturated and unsaturated porous media: Role of ionic strength, pH, and collector grain size. Water research (2018) 133, 338-347.

  1. Another technique that can be sued to characterise CDs is Raman microscopy and can give important information on the disorder or defective nature of the CDs. This is particularly relevant before and after functionalisation. Some mention of this technique and its potential in characterising these types of materials would be prudent to mention. The following paper is an example where Raman was applied to CD and also shows an interesting application for CD nanocomposites in fingermark detection.

Nontoxic luminescent carbon dot/poly (dimethylacrylamide) nanocomposite reagent for latent fingermark detection synthesized via surface initiated reversible addition fragmentation chain transfer polymerization, J Dilag, H Kobus, Y Yu, CT Gibson, AV Ellis, Polymer International 64 (7), 884-891, 2015.  

 Our response: We thank the reviewer for pointing the use of RAMAN spectroscopy for the characterization of CD defects. However, the presented data using FT-IR provide clear evidence on the functionalization of the CDs. We do not think that such additional data will significantly alter the content of the study or the interpretation of the results. Additionally, Data from RAMAN spectroscopy were not collected and therefore cannot be provided at this point. No changes were made in response to this comment.

Reviewer 2 Report

The authors submitted "Transport of N-CD and pre-sorbed Pb in saturated
porous media" to publish in Molecules. Here my comments:

  1. 4 gr should be 4 g.
  2. Introduction part is short. 
  3. the authors repeat two time about Characterization of N-CDs. please ckarify.
  4. Figure 1 is more crowded. please modify it.
  5. the authors used Fourier transform-infrared (FT-IR) measurements to confirm the structure of N-CD and nonfunctionalized CDs. are there other instruments to confirm thier structure?
  6. English correction is needed.

Author Response

Dear Editor,

We thank the reviewers for their constructive feedback, which helped us to improve the manuscript. Below, we address the reviewers’ comments one by one and discuss the subsequent modifications to the manuscript. All requested changes have been marked in blue.

Reviewer 2

Comments and Suggestions for Authors

The authors submitted "Transport of N-CD and pre-sorbed Pb in saturated porous media" to publish in Molecules. Here my comments:

  1. 4 gr should be 4 g.

Our response: The correction was done (page number 2, section 2.1, line 3)

  1. Introduction part is short. 

Our response: Introduction was extended (page number 2) “…. Li et al. (Li et al. 2019) compared the transport behavior of N-doped graphene (NG) and Graphene oxide (GOs) in porous media, and demonstrated that GOs were more mobile than NG in quartz column and that the mobility of NG decreased with increases in ionic strength and decreases in quartz media grain size. Additionally, the facilitated transport of heavy metals by carbon-based nanoparticles have received significant attention recently. For instance, Zhou et al. (Zhou et al. 2016) investigated the effects of GO concentration and IS on the transport of Cu2+ in the sand column. They reported increases in Cu2+ transport with decreases in IS and increases in GO concentrations. Additionally, GO have been shown to remobilize pre-sorbed Pb2+ and Cd2+ in porous media (Yin et al. 2019). ….”. See page 2, 2nd paragraph.

  1. The authors repeat two time about Characterization of N-CDs. please ckarify.

Our response: It is not clear what does the reviewer refer to here. Nonetheless, we changed the second mention of “Characterization of N-CDs” in the manuscript to “Properties of N-CDs”. See page 4.

  1. Figure 1 is more crowded. please modify it.

Our response: Figure 1 was modified as shown below to reduce crowdedness (See page 5)

  1. The authors used Fourier transform-infrared (FT-IR) measurements to confirm the structure of N-CD and nonfunctionalized CDs. are there other instruments to confirm their structure?

Our response: The CDs were characterized by DLS, FT-IR and UV-Vis spectroscopy (presented in this study) and by TEM (published in previous studies, see reference below). We note that the aim of this study was not to investigate the detailed structure of N-CDs, but rather to investigate the interaction of Pb with CDs and quartz and the transport of CDs and Pb in quartz columns. We understand that the more characterization of the CDs the better, but we believe that the presented characterization (e.g., size, surface charge, and surface functional groups) is sufficient to underpin the findings of this study. No changes were made in response to this comment.

Reference: Kamrani S, Rezaei S, Kord M, Baalousha M. Transport and retention of carbon dots (CDs) in saturated and unsaturated porous media: Role of ionic strength, pH, and collector grain size. Water research (2018) 133, 338-347.

  1. English correction is needed.

Our response: The manuscript was proof-read again and English correction was made throughout the manuscript (in red/blue).

Round 2

Reviewer 2 Report

This manuscript should be accepted.

Author Response

Reviewer 2 (Round 2)

Comments and Suggestions for Authors

This manuscript should be accepted.

Our response: We thank the reviewer for the positive feedback.

Academic Editor Notes

I think that the authors should supplement the Raman analysis according to the suggestion of the reviewer 1, because Raman spectrum may identify the ordered or disordered structure of CDs.

Our response: Unfortunately we don’t have access to the RAMAN spectroscopy. We also believe that such data will not impact/change the conclusion of the study.